# AGENTCHANGEBENCH: A MULTI-DIMENSIONAL EVALUATION FRAMEWORK FOR GOAL-SHIFT ROBUSTNESS IN CONVERSATIONAL AI

## ABSTRACT

Goal changes are a defining feature of real world multi-turn interactions, yet current agent benchmarks primarily evaluate static objectives or one-shot tool use. We introduce **AgentChangeBench**, a benchmark explicitly designed to measure how tool augmented language model agents adapt to mid dialogue goal shifts across four enterprise domains. Our framework formalizes evaluation through four complementary metrics: Task Success Rate (TSR) for effectiveness, Tool Use Efficiency (TUE) for reliability, Tool Call Redundancy Rate (TCRR) for wasted effort, and Goal-Shift Recovery Time (GSRT) for adaptation latency. AgentChangeBench comprises of 590 task sequences and five user personas, each designed to trigger realistic shift points in ongoing workflows. Using this setup, we evaluate a mix of proprietary and open source models and uncover sharp contrasts obscured by traditional pass@k scores. Our findings demonstrate that high raw accuracy does not imply robustness under dynamic goals, and that explicit measurement of recovery time and redundancy is essential. AgentChangeBench establishes a reproducible testbed for diagnosing and improving agent resilience in realistic enterprise settings.

## 1 INTRODUCTION

Large Language Models (LLMs) have rapidly advanced as conversational agents capable of reasoning, tool use, and multi-turn interaction across diverse domains. However, most existing benchmarks for evaluating LLM-as-agent performance assume that user goals remain fixed throughout a conversation. This assumption oversimplifies real-world deployments, where users frequently re-prioritize tasks, introduce new constraints, or shift objectives mid-dialogue. For example, a banking customer may begin by authenticating their identity, then pivot to reviewing transactions, and finally escalate to disputing a fraudulent charge, all within the same interaction. Evaluating agent robustness in such dynamic contexts is critical for enterprise adoption of LLM-based assistants.

To address this gap, we introduce AgentChangeBench, a comprehensive evaluation framework that systematically measures how well conversational agents detect, adapt, and recover from multi-turn changes in user objectives, as well as how they tailor their instructional strategies to diverse user personas with varying levels of expertise, cooperation, and trust. Our work builds upon advances in persona-based user simulation (Li et al., 2016; Zhang et al., 2018; Schatzmann et al., 2007) and systematic benchmark creation (Liang et al., 2022; Srivastava et al., 2022), while extending evaluation to dynamic goal shift scenarios.

Our contributions are threefold:

1. **Novel evaluation focus:** We design the first benchmark explicitly testing how LLM agents handle mid-conversation goal shifts and adapt communication for diverse user personas

2. **Comprehensive coverage:** We provide 590 systematically validated tasks across four domains (banking, retail, airline, education) with five personas and explicit goal shifts

3. **Methodological framework:** We introduce evaluation protocols for goal shift recovery designed for realistic customer personas, improving the scope of multi-turn LLM assessment

Table 1: Comparison on goal dynamics, persona coverage, and tool evaluation.

| Benchmark | Goal dynamics | Personas | Tool use |
|---|---|---|---|
| $\tau$-bench | Static objectives | Limited | Domain APIs |
| $\tau^2$-bench | Mostly static | Several | Domain APIs |
| AgentBench | Task-defined (stable) | None | Varied tools |
| **This work** | **Explicit goal sequences** | **Five** | Domain APIs |

4. **Empirical study:** We run a cross-model evaluation that reveals significant divergences among state-of-the-art models in success, recovery time, efficiency, and redundancy, surfacing trade-offs that pass$^k$ alone does not capture.

To contextualize our contributions within the existing literature and highlight the novelty of our approach, we first review related work in conversational AI evaluation, with particular focus on benchmarks that address tool use and multi-turn interactions.

**Release.** To support reproducibility, we have released the full benchmark, evaluation harness configurations, along with all experimental artifacts as supplementary material with our paper.

## 2 RELATED WORK

$\tau$-bench (Yao et al., 2024) introduced simulated multi-turn interactions in retail and airline contexts, emphasizing API tool usage and providing the pass$^k$ metric for measuring consistency across runs. While effective for tool-centric evaluation, $\tau$-bench assumes static user goals and full agent control over the environment, limiting its ability to capture dynamic conversational shifts. $\tau^2$-bench (Barres et al., 2025) extended this line of work by modeling telecom support scenarios requiring user-agent coordination. It introduced compositional task generation but remained restricted to a narrow set of personas and contexts, without testing adaptability to changing user goals or runtime constraints.

More open-ended benchmarks such as AgentBench (Liu et al., 2024) evaluate LLM-as-Agent capabilities across eight interactive environments such as operating systems, databases, and web browsing. Although it broadens domains beyond traditional customer service, AgentBench similarly evaluates agents under stable user objectives, leaving open the question of how agents behave under dynamically shifting goals or varied communication demands. Recent work has begun exploring adaptive conversation flows, but focuses primarily on single-domain interactions without the multi-domain goal-shift scenarios we address.

Taken together, these efforts provide strong foundations for tool-use and multi-turn evaluation, but they differ in how they address (or neglect) shifting goals and persona diversity. Table 1 highlights this contrast using the notion of *explicit goal sequences* rather than fixed goals. Each task specifies an ordered sequence of goals, the persona-conditioned user simulator enacts the corresponding shifts, and the evaluator computes Goal Shift Recovery Time from the transcript (acknowledgment, tool, outcome), reported alongside TSR, TUE, and TCRR.

## 3 METHODOLOGY

### 3.1 DATASET DESIGN

We construct a benchmark of 590 curated multi-turn tasks across four domains (banking: 100 tasks, airline: 150 tasks, retail: 215 tasks, education: 100 tasks) grounded in real-world customer service workflows. Each domain incorporates realistic goal transitions with five distinct user personas and explicit goal shifts. Our domain selection aligns with common customer-service workflows across financial services, retail omnichannel, and airline support.

Table 2: Five user personas with distinct conversational styles and task coverage.

| Persona | Characteristics | Interaction style |
|---|---|---|
| EASY_1 | Polite, detail-oriented, step-by-step | "Please walk me through..." |
| EASY_2 | Easily distracted, casual, confused | "Oh wait, actually..." |
| MEDIUM_1 | Business-focused, impatient, efficient | "I need this done quickly" |
| MEDIUM_2 | Curious learner, asks questions | "Can you teach me about..." |
| HARD_1 | Suspicious, questioning, demands proof | "How do I know this is secure?" |

## 3.2 TASK GENERATION

Our benchmark construction began with hand-converted exemplars designed to capture realistic workflows, conversational turns, and domain-specific constraints. We seed many retail and airline scenarios from $\tau^2$: we reuse 50 airline and 114 retail templates, and contribute **50** newly generated scenarios (in both airline and retail). Banking and education coverage is entirely original (200 tasks). Across all four domains, we add explicit goal-sequence annotations, broaden persona coverage, and enforce uniform shift-triggering rules.

In total, the dataset comprises **590** tasks spanning banking (**100**), airline (**150**), education (**100**),and retail (**215**). Each task specifies one of five personas and an explicit ordered list of goals (e.g., `["authentication","transactions","dispute"]`).

## 3.3 USER PERSONAS

To simulate realistic conversational variation, we defined five personas with distinct behavioral traits, interaction styles, and levels of cooperation. Each persona was allocated tasks proportionally, ensuring balanced coverage across the dataset. The distribution of personas among tasks can be found in Table 4.

## 3.4 TASK SCHEMA

Tasks follow a declarative JSON schema specifying persona, known and unknown information, and an ordered list of goals. Each task declares a `goal_shifts` object of the form:

```
"goal_shifts": { "required_shifts": k,
                 "goals": ["g1","g2",...,"g{k+1}"] }
```

where `required_shifts = len(goals)-1`. Transitions are triggered naturally (e.g., after four user turns on the same goal, after a helpful resolution step, or when the agent asks "anything else?"). Agents never see markers.

**Examples.** We instantiate $>150$ unique goal labels spanning airline (`reservation`, `baggage`, `cancellation`), retail (`returns`, `exchange`, `order_tracking`), and banking (`statements`, `fraud_response`, `payments`). Tasks range from single-goal flows (`["payments"]`) to more complex sequences such as `["authentication","transactions","dispute"]` or `["insights","fraud_response"]`.

**User/agent control.** Across banking, retail, education, and airline, the user never issues tool calls. They only disclose facts already present in the task's `known_info` (e.g., name, phone, order/booking IDs), while the assistant performs all tool interactions (e.g., `unlock_card`, `return_delivered_order_items`, `update_reservation_flights`).

## 3.5 EVALUATION HARNESS

We employed the $\tau^2$-bench evaluation harness as the backbone of our experimental setup. The harness provided a controlled environment for executing our tasks, enforcing constraints such as

Table 3: Dataset comparison across conversational AI benchmarks.

| Benchmark | Tasks | Domains | Personas | Goal shifts | Metrics |
|---|---|---|---|---|---|
| $\tau$-bench | 234 | 2 | 3 | None | pass$^k$ only |
| $\tau^2$-bench | 105 | 1 | 5 | Implicit | pass$^k$ + modes |
| AgentChangeBench | 590 | 4 | 5 | Explicit | Multi-dimensional |

one-tool-per-turn, policy adherence, and correct sequencing of goal shifts. This allowed us to systematically test how agents re-plan when confronted with mid-dialogue goal changes and whether they adjust communication strategies to match user personas.

### 3.6 ANALYSIS

#### 3.6.1 DATASET QUALITY ANALYSIS

We conduct comprehensive analysis of our dataset quality compared to existing benchmarks, demonstrating the enhanced coverage and evaluation capabilities of AgentChangeBench.

**Task Coverage and Diversity.** Our dataset comprises 590 tasks across four domains, significantly expanding upon $\tau$-bench's 234 tasks and $\tau^2$-bench's 105 tasks. Table 3 provides detailed comparison.

### 3.7 COMPARATIVE ANALYSIS AND INSIGHTS

Our evaluation framework reveals performance characteristics that traditional binary metrics miss. For instance, agents with similar pass$^k$ scores can exhibit dramatically different TUE and TCRR values, indicating varying levels of operational efficiency. Similarly, GSRT analysis shows that some agents achieve similar final success rates but require significantly different recovery times under goal shifts.

This granular analysis enables more informed deployment decisions. Organizations can optimize agents for specific scenarios: financial services companies might prioritize TUE and TCRR for cost control, while customer service organizations might emphasize GSRT and communication quality for user experience.

**Summary.** Together, these metrics evaluate four complementary dimensions of performance necessary for agent deployment in dynamic, enterprise-grade conversational settings: (1) Can the agent succeed? (TSR), (2) How efficiently does it use tools? (TUE), (3) Does it avoid waste? (TCRR), and (4) Can it adapt quickly under evolving user goals? (GSRT and retention/drop analysis). By combining efficiency, redundancy, and recovery time across multiple dimensions, our framework advances beyond prior benchmarks, offering a more realistic and actionable view of agent performance in dynamic multi-turn conversations.

Having demonstrated the effectiveness and comprehensiveness of our evaluation framework through extensive experimentation and analysis, we now summarize our contributions and discuss their implications for the future of conversational AI evaluation.

## 4 LIMITATIONS AND FUTURE WORK

**Persona difficulty and coverage.** Our five personas vary tone and cooperation, but they are still relatively benign. They do not yet stress adversarial, deceptive, hostile, or policy-pushing behaviors, and they rarely force long-horizon memory or multi-goal juggling. We plan to add *hard* personas (e.g., adversarial or non-cooperative users, conflicting instructions, frequent interruptions, implicit constraints, multilingual switches) to better probe boundary cases and safety.

**Domain and tool scope.** AgentChangeBench currently focuses on customer-service style workflows (banking, retail, airline) with domain APIs. We do *not* include other important tool classes such as IDE/code-editor actions, OS/shell control, spreadsheet/BI tools, browsers, or robotics/IoT

controllers, and the harness does not yet use a unified tool protocol (e.g., MCP). Future releases will broaden coverage to these tool types and provide MCP-compatible adapters so agents can operate across heterogeneous tools with a single interface.

**Goal-shift specification.** Goal shifts are pre-declared sequences executed by the user simulator and are often explicitly signaled. We do not yet evaluate detection of *implicit* goal drift, overlapping/interleaved objectives, or conflicts between goals. We will introduce latent and ambiguous shifts, partial reversions, and concurrent subgoals to test plan repair under uncertainty.

**Model and coverage breadth.** We evaluate three major model families on four domains. Expanding to more model sizes and architectures (including open-weight models) and to additional domains (e.g., healthcare, education, technical support) will improve generality.

**Summary.** Despite these constraints, AgentChangeBench surfaces adaptation and efficiency gaps that success-only metrics miss. Broadening personas, tools (including code/OS tools), protocol support (MCP), and evaluation settings is a direct path to harder, more realistic benchmarks.

## 5 CONCLUSION

We introduced AgentChangeBench, a benchmark for evaluating conversational agents under dynamic goal shifts. Our 590 tasks span banking, retail, education, and airline domains with five distinct personas, each annotated with explicit goal sequences. Beyond binary success, we propose four complementary metrics (TSR, TUE, TCRR, and GSRT) that capture success, efficiency, redundancy, and recovery.

Experiments across three major LLM families highlight clear differences in robustness and adaptation: Claude-3.7-Sonnet recovers fastest, GPT-4o delivers balanced cross-domain performance, and Gemini-2.5-Flash lags in banking but remains competitive in retail. These results demonstrate the need for multi-dimensional evaluation to surface tradeoffs that $pass^k$ alone cannot reveal.

Future work can extend AgentChangeBench to new domains as well as develop methods for automated task generation. Another promising direction would be incorporating multilingual settings, to better capture the challenges of human–AI interaction in realistic settings.

We thank the $\tau$-bench and $\tau^2$-bench teams for releasing their tasks and evaluation harness, which served as the foundation for our extensions with explicit goal shifts and persona coverage. We also acknowledge the MultiWOZ community for prior benchmarks that informed our design choices.

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

## A  APPENDIX

## B  DATASET AND TASK DETAILS

### B.1  TASK GENERATION METHODOLOGY

Our task generation process follows a systematic approach to ensure comprehensive coverage across domains and personas. Each domain undergoes a five-stage development process:

**Stage 1: Domain Analysis.** We analyze real-world customer service scenarios to identify common user intents, required tools, and typical conversation flows. This analysis forms the foundation for task design.

**Stage 2: Tool Definition.** Based on domain analysis, we define a comprehensive set of tools that agents can use to accomplish user goals. Tools are designed to reflect real-world APIs and capabilities.

**Stage 3: Task Template Creation.** We create task templates that combine user scenarios, required tools, and evaluation criteria. Each template specifies the initial state, user goal, required actions, and success conditions.

**Stage 4: Persona Integration.** Tasks are enhanced with persona-specific variations that reflect different user characteristics, technical expertise levels, and interaction patterns.

**Stage 5: Goal Shift Integration.** We systematically introduce goal shifts with realistic adaptation scenarios, ensuring challenging but achievable goal transitions.

### B.2  EXAMPLE TASK

```
1  Task ID: 10_banking_cards_medium_1_dispute_001
2  Description:
3  - Purpose: Card unlock request to dispute filing - MEDIUM_1 persona (
       business-focused)
4  - Relevant Policies: Security protocols before unlock; dispute handling
5  User Scenario:
6  - Persona: MEDIUM_1
7  - Domain: banking
8  - Reason for Call: Unlock card, then file dispute for unauthorized charge
9  Known Information:
10 - Name: Taylor Johnson
11 - Phone: +15551230987
12 - Date of Birth: 1991-05-06
13 - Email: user.003@example.com
14 - Unauthorized Transaction: $149.99 at 'SUSPICIOUS MERCHANT 123' on
       2025-06-18 at 16:25 (Transaction ID: tx_303)
15 Unknown Information:
16 - Dispute process details and resolution timeline
17 Task Instructions:
18 1. Request to unlock your card
```

```
19  2. File a dispute for the unauthorized transaction (tx_303) for $149.99
        at 'SUSPICIOUS MERCHANT 123' from 2025-06-18
20  Goal Shifts:
21  - Required Shifts: 1
22  - Goals: ["cards", "dispute"]
23  Initial State:
24  - Phone Number: +15551230987
25  - Customer ID: cust_303
26  - Primary Card ID: card_303
27  - Primary Card Active: false
28  - Primary Account ID: acc_303
29  Evaluation Criteria:
30  Action Sets:
31  1. verify_identity
32     - Allowed Tools: get_customer_by_phone, get_customer_by_id
33     - Max Score: 1.0
34     - Scoring: parameter_accuracy (1.0), tool_usage (1.0)
35  2. unlock_card_request
36     - Allowed Tools: unlock_card
37     - Max Score: 1.0
38     - Scoring: parameter_accuracy (1.0), tool_usage (1.0)
39  Natural Language Assertions:
40  - Agent verified customer identity before processing card unlock
41  - Agent clearly explained the card unlock process and timing
42  - Agent guided through the dispute filing process for the unauthorized
        transaction
43  - Agent did not transfer the customer to a human agent when the goal
        changed
44  Communication Information:
45  - $149.99
46  - acc_303
47  - tx_303
```

Listing 1: Banking Card Task Configuration

```
1   Task ID: 10_banking_cards_medium_1_dispute_001
2
3   Description:
4   - Purpose: Card unlock request to dispute filing – MEDIUM_1 persona (
        business-focused)
5   - Relevant Policies: Security protocols before unlock; dispute handling
6
7   User Scenario:
8   - Persona: MEDIUM_1
9   - Domain: banking
10  - Reason for Call: Unlock card, then file dispute for unauthorized charge
11
12  Known Information:
13  - Name: Taylor Johnson
14  - Phone: +15551230987
15  - Date of Birth: 1991-05-06
16  - Email: user.003@example.com
17  - Unauthorized Transaction: $149.99 at 'SUSPICIOUS MERCHANT 123' on
        2025-06-18 at 16:25 (Transaction ID: tx_303)
18
19  Unknown Information:
20  - Dispute process details and resolution timeline
21
22  Task Instructions:
23  1. Request to unlock your card
24  2. File a dispute for the unauthorized transaction (tx_303) for $149.99
        at 'SUSPICIOUS MERCHANT 123' from 2025-06-18
25
26  Goal Shifts:
27  - Required Shifts: 1
```

```
28  - Goals: ["cards", "dispute"]
29
30  Initial State:
31  - Phone Number: +15551230987
32  - Customer ID: cust_303
33  - Primary Card ID: card_303
34  - Primary Card Active: false
35  - Primary Account ID: acc_303
36
37  Evaluation Criteria:
38
39  Action Sets:
40  1. verify_identity
41     - Allowed Tools: get_customer_by_phone, get_customer_by_id
42     - Max Score: 1.0
43     - Scoring: parameter_accuracy (1.0), tool_usage (1.0)
44
45  2. unlock_card_request
46     - Allowed Tools: unlock_card
47     - Max Score: 1.0
48     - Scoring: parameter_accuracy (1.0), tool_usage (1.0)
49
50  Natural Language Assertions:
51  - Agent verified customer identity before processing card unlock
52  - Agent clearly explained the card unlock process and timing
53  - Agent guided through the dispute filing process for the unauthorized
       transaction
54  - Agent did not transfer the customer to a human agent when the goal
       changed
55
56  Communication Information:
57  - $149.99
58  - acc_303
59  - tx_303
```

Listing 2: Banking Card Task Configuration

## B.3 PERSONA DEFINITIONS

```
1  # EASY_1
2
3  **Personality & Tone:** Patient, friendly, casual. Takes time to
      understand options and doesn't rush decisions. Appreciates
      explanations and guidance.
4
5  **Speaking Style:**
6  - Conversational and polite: "Hi there!" "Thanks so much!" "I appreciate
      your help"
7  - Patient with processes: "No rush" "I have time" "Whatever works best"
8  - Asks clarifying questions: "What does that mean?" "Could you explain
      that?"
9  - Expresses gratitude: "You've been so helpful" "Thank you for your
      patience"
10
11  **Expertise:** Low travel experience; needs guidance on airline policies,
      baggage rules, and booking processes. Often asks basic questions
      about flights and procedures.
12
13  **Technology Comfort:** Medium; comfortable with basic online
      interactions but may need help with complex processes like seat
      selection or payment methods.
14
```

```
15  **Goal-Change Behavior:** Gradual transitions with clear explanations.
        Uses phrases like "Oh, I just thought of something else" "While I
        have you on the line" "Actually, I also need to..."
16
17  **Common Phrases:**
18  - "I'm not really sure how this works"
19  - "Is that the best option for me?"
20  - "What would you recommend?"
21  - "I want to make sure I understand"
22
23  # EASY_2
24
25  **Personality & Tone:** Warm, family-focused, detail-oriented. Concerned
        about everyone's needs and comfort. Wants to ensure everything goes
        smoothly for the family.
26
27  **Speaking Style:**
28  - Family-centered: "For my family" "My kids" "My husband and I" "We're
        traveling with children"
29  - Detail-focused: "Let me make sure I have this right" "What about...?" "
        I need to double-check"
30  - Accommodating: "Whatever works for everyone" "Is this family-friendly?"
         "Can we sit together?"
31  - Practical: "What's the most convenient option?" "How does this work
        with kids?"
32
33  **Expertise:** Moderate; understands basic travel but asks about family-
        specific policies, child discounts, and group bookings.
34
35  **Technology Comfort:** Medium; comfortable with standard booking but may
         need help with multiple passengers or special requests.
36
37  **Goal-Change Behavior:** Transitions based on family needs discovery.
        Uses phrases like "Oh, I forgot about the kids" "My spouse just
        reminded me" "For the family trip, we also need..."
38
39  **Common Phrases:**
40  - "We're traveling as a family"
41  - "What's best for traveling with children?"
42  - "I need to coordinate for everyone"
43  - "Is there a family discount?"
44
45  # MEDIUM_1
46
47  **Personality & Tone:** Direct, efficient, professional. Time-conscious
        and expects streamlined service. Familiar with travel processes but
        focused on business needs.
48
49  **Speaking Style:**
50  - Professional and direct: "I need to..." "Can you..." "What's the
        timeline?"
51  - Time-conscious: "I'm on a tight schedule" "How quickly can this be done
        ?" "Time is important"
52  - Business-focused: "For business travel" "Company policy requires" "I
        need flexibility"
53  - Solution-oriented: "What are my options?" "What's the best approach?" "
        How do we fix this?"
54
55  **Expertise:** High; understands airline policies, loyalty programs, and
        business travel requirements. Uses industry terminology confidently.
56
57  **Technology Comfort:** High; expects efficient digital processes and
        self-service options when possible.
58
```

```
59  **Goal-Change Behavior:** Efficient stacking of requests. Uses phrases
        like "While we're at it" "I also need to handle" "Can we take care of
         multiple items?"
60
61  **Common Phrases:**
62  - "This is for business travel"
63  - "I need flexible options"
64  - "What's the most efficient way?"
65  - "I travel frequently"
66
67  # MEDIUM_2
68
69  **Personality & Tone:** Practical, cost-aware, research-oriented.
        Compares options carefully and seeks the best value. Willing to trade
         convenience for savings.
70
71  **Speaking Style:**
72  - Cost-focused: "What's the cheapest option?" "Are there any fees?" "How
        much would that cost?"
73  - Comparison-oriented: "What's the difference between...?" "Which is
        better value?" "Are there alternatives?"
74  - Practical: "I don't need all the extras" "Basic is fine" "What's
        included?"
75  - Research-minded: "I've been looking at options" "I saw online that..."
        "Can you match this price?"
76
77  **Expertise:** Medium-High; knowledgeable about finding deals, airline
        policies, and hidden fees. Asks detailed questions about costs.
78
79  **Technology Comfort:** High; comfortable comparing options online and
        using price comparison tools.
80
81  **Goal-Change Behavior:** Discovers additional cost-saving opportunities
        during conversation. Uses phrases like "Speaking of saving money" "
        While I'm optimizing costs" "Are there other ways to save?"
82
83  **Common Phrases:**
84  - "What's the most affordable option?"
85  - "Are there any hidden fees?"
86  - "I'm trying to stick to a budget"
87  - "Is there a better deal?"
88
89  # HARD_1
90
91  **Personality & Tone:** Nervous, cautious, seeks reassurance. Worried
        about making mistakes or missing important details. Needs extra
        support and patience.
92
93  **Speaking Style:**
94  - Anxious and questioning: "I'm worried that..." "What if...?" "Am I
        doing this right?"
95  - Seeks reassurance: "Is that normal?" "Are you sure?" "What should I
        expect?"
96  - Detail-concerned: "I don't want to mess this up" "Can you double-check
        ?" "What happens if...?"
97  - Apologetic: "Sorry for all the questions" "I'm new to this" "I hope I'm
         not bothering you"
98
99  **Expertise:** Very low; unfamiliar with airline procedures, policies,
        and travel requirements. Needs step-by-step guidance.
100
101 **Technology Comfort:** Low-Medium; may struggle with complex online
        processes and prefer human assistance.
102
```

```
103  **Goal-Change Behavior:** Reactive to new concerns that arise during
         conversation. Uses phrases like "Oh no, I just realized" "Wait, what
         about...?" "I'm suddenly worried about..."
104
105  **Common Phrases:**
106  - "I've never flown before"
107  - "I'm really nervous about this"
108  - "Can you walk me through this?"
109  - "What do I need to know?"
```

## B.4 EXAMPLE TASK RESULT

Result of running the example task in Appendix A.2 with GPT-4o.

```
1   Assistant: Hi! How can I help you today?
2
3   User: Unlock my card, then file a dispute for an unauthorized charge.
4
5   Assistant: To proceed with unlocking your card and filing a dispute for
         an unauthorized charge, I need to verify your identity and gather
         some details.
6
7   1. **Unlock Card**: Please confirm the card ID you want to unlock.
8   2. **File Dispute**: Provide the account ID and transaction ID for the
         unauthorized charge, along with the reason code "unauthorized."
9
10  Once I have this information, I'll provide a summary of the actions for
         your confirmation.
11
12  User: I already gave the transaction details. Check the notes. My card is
          linked to my account, so just unlock it. Transaction ID for the
         dispute is tx_303, $149.99 at 'SUSPICIOUS MERCHANT 123' on
         2025-06-18.
13
14  Assistant: Before proceeding, I need to verify your identity. Could you
         please provide your customer ID or the full registered phone number
         associated with your account?
15
16  User: Phone number is +15551230987. Let's get this moving.
17
18  Assistant: [Tool call: get_customer_by_phone]
19
20  Tool: [Tool call: get_customer_by_phone(phone_number: "+15551230987")]
21  {
22    "customer_id": "cust_303",
23    "full_name": "Taylor Johnson",
24    "date_of_birth": "1991-05-06",
25    "email": "user.003@example.com",
26    "phone_number": "+15551230987",
27    "address": {
28      "street": "321 Maple Drive",
29      "city": "San Diego",
30      "state": "CA",
31      "postal_code": "90004"
32    },
33    "created_at": "2025-01-01 00:00:00",
34    "account_ids": ["acc_303"],
35    "card_ids": ["card_303"],
36    "statement_ids": [],
37    "payment_request_ids": [],
38    "dispute_ids": [],
39    "payee_ids": ["payee_303"]
40  }
41
```

```
42  Assistant: Thank you for providing your information. Here are the actions
        I will take:
43
44  1. **Unlock Card**: Unlock the card with ID 'card_303'.
45  2. **File Dispute**: File a dispute for the transaction with ID 'tx_303'
        for $149.99 at 'SUSPICIOUS MERCHANT 123' on 2025-06-18, with the
        reason code "unauthorized."
46
47  Please confirm with "YES" to proceed with these actions.
48
49  User: YES. Confirm both actions.
50
51  Assistant: [Tool call: unlock_card]
52
53  Tool: [Tool call: unlock_card(card_id: "card_303")]
54  {
55    "card_id": "card_303",
56    "status": "Active"
57  }
58
59  Assistant: [Tool call: file_dispute]
60
61  Tool: [Tool call: file_dispute(account_id: "acc_303", tx_id: "tx_303",
        reason_code: "unauthorized")]
62  Error: DISPUTED
63
64  Assistant: The card with ID 'card_303' has been successfully unlocked.
65
66  However, there was an error filing the dispute for the transaction with
        ID 'tx_303'. It seems the transaction might already be disputed or
        there is another issue.
67
68  Would you like me to transfer you to a human agent for further assistance
         with the dispute?
69
70  User: Before the transfer, I also wanted to ask about the wire transfer
        limits. Can you provide that information?
71
72  Assistant: I can provide information about wire transfer limits. Please
        hold on while I retrieve the details for your account.
73
74  Tool: [Tool call: get_account(account_id: "acc_303")]
75  {
76    "account_id": "acc_303",
77    "customer_id": "cust_303",
78    "type": "checking",
79    "masked_number": "        3456        ",
80    "status": "Active",
81    "current_balance": 1850.25,
82    "available_balance": 1800.0
83  }
84
85  Assistant: I have retrieved your account details, but it seems I need to
        access specific information about wire transfer limits. Unfortunately
        , I don't have direct access to this information.
86
87  I will now transfer you to a human agent who can assist you further with
        both the dispute issue and the wire transfer limits. Please hold on.
88
89  Tool: [Tool call: transfer_to_human_agents(summary: "The user needs
        assistance with filing a dispute for a transaction that encountered
        an error and requires information on wire transfer limits for their
        account.")]
90  Transfer successful
91
```

```
92  Assistant: YOU ARE BEING TRANSFERRED TO A HUMAN AGENT. PLEASE HOLD ON.
93
94  User: ###TRANSFER###
```

Listing 3: Example Task Result

## B.5 GOAL SHIFT CONFIGURATION DETAILS

**Definition.**

```
"goal_shifts": {
  "required_shifts": k,
  "goals": ["g1","g2",...,"g{k+1}"]
}
```

**Semantics.**

- Goals are executed in order by the user simulator; agents never see markers.
- `required_shifts` must equal `len(goals)-1` (validator enforced).
- Shifts are triggered by natural conversation flow and progression rules.

**Examples.**

```
["authentication","transactions","dispute"]  // banking
["payments","account_info"] // retail/airline (/ˆ2 adaptations)
```

**User model (all domains).** Users expose only `known_info`; *all* tool use is by the assistant. Progression rules prevent stalls and trigger natural transitions.

## C  EVALUATION METHODOLOGY

### C.1  DETAILED METRIC CALCULATIONS

**TSR Component Weights:** The weights used in TSR calculation are determined through empirical analysis of task importance:

- communicate_info: 0.25 (25% weight)
- action: 0.45 (45% weight)
- nl_assertion: 0.30 (30% weight)

These weights reflect the relative importance of each component in determining overall task success.

**TUE Component Weights:** Tool Usage Efficiency weights are based on operational cost analysis:

- tool_correctness: 0.6 (60% weight)
- param_accuracy: 0.4 (40% weight)

The higher weight for tool correctness reflects its critical importance in successful task execution.

**TCRR Parameters:**

- window_size: 3 turns
- batch_threshold: 2 calls

These parameters are optimized to detect both cross-turn duplicates and intra-turn batch inefficiencies.

## C.2 EVALUATION PROTOCOL

**Simulation Setup:**

- Each task is evaluated across 3 independent runs
- User simulator follows persona-specific behavior patterns
- Environment state is reset between runs
- Tool calls are validated against actual API responses

**Scoring Process:** 1. Task execution is monitored for all required components 2. Tool calls are validated for correctness and parameter accuracy 3. Communication quality is assessed against required information 4. Behavioral compliance is evaluated through natural language assertions 5. Goal shift recovery is measured across all adaptation scenarios

**Quality Assurance:** - Manual review of 10% of tasks for validation - Cross-checking of evaluation criteria consistency - Statistical analysis of inter-rater reliability - Regular updates based on feedback and edge cases

## D IMPLEMENTATION DETAILS

### D.1 TOOL DEFINITIONS

**Banking Tools:**

```
get_customer_by_id(customer_id)
get_customer_by_phone(phone_number)
get_customer_by_name(full_name, dob)
get_accounts(customer_id)
get_account(account_id)
get_statements(account_id, limit)
get_transactions(account_id, start_time, end_time, limit)
add_payee(customer_id, name, deliver_type)
create_payment_request(
    customer_id,
    from_account_id,
    to_payee_id,
    amount,
    expires_at
)
check_payment_request(request_id)
authorize_payment_request(request_id)
make_payment(request_id)
cancel_payment_request(request_id)
lock_card(card_id, reason)
unlock_card(card_id)
file_dispute(account_id, tx_id, reason_code)
get_dispute(dispute_id)
park_task(current_task_id, resume_hint)
resume_task(parked_task_id)
transfer_to_human_agents(summary)
```

**Retail Tools:**

```
calculate(expression)
cancel_pending_order(order_id, reason)
exchange_delivered_order_items(
    order_id,
    item_ids,
```

```
        new_item_ids,
        payment_method_id
    )
    find_user_id_by_name_zip(first_name, last_name, zip)
    find_user_id_by_email(email)
    get_order_details(order_id)
    get_product_details(product_id)
    get_user_details(user_id)
    list_all_product_types()
    modify_pending_order_address(
        order_id,
        address1,
        address2,
        city,
        state,
        country,
        zip
    )
    modify_pending_order_items(order_id, item_ids, new_item_ids, payment_method_id)
    modify_pending_order_payment(order_id, payment_method_id)
    modify_user_address(user_id, address1, address2, city, state, country, zip)
    return_delivered_order_items(order_id, item_ids, payment_method_id)
    transfer_to_human_agents(summary)

    Airline Tools:

    book_reservation(
        user_id,
        origin,
        destination,
        flight_type,
        cabin,
        flights,
        passengers,
        payment_methods,
        total_baggages,
        nonfree_baggages,
        insurance
    )
    calculate(expression)
    cancel_reservation(reservation_id)
    get_reservation_details(reservation_id)
    get_user_details(user_id)
    list_all_airports()
    search_direct_flight(origin, destination, date)
    search_onestop_flight(origin, destination, date)
    send_certificate(user_id, amount)
    transfer_to_human_agents(summary)
    update_reservation_baggages(
        reservation_id,
        total_baggages,
        nonfree_baggages,
        payment_id
    )
    update_reservation_flights(reservation_id, cabin, flights, payment_id)
    update_reservation_passengers(reservation_id, passengers)
    get_flight_status(flight_number, date)
```

# E    ADDITIONAL RESULTS

## E.1    OVERVIEW

This appendix reports full per-model, per-domain results beyond the compact summaries in the main text. We focus on (i) overall task effectiveness (TSR and its channel components), (ii) operational efficiency (TUE), (iii) redundancy (TCRR), and (iv) adaptation under goal shifts (GSRT). Three consistent patterns emerge across models and domains: (1) *Redundancy dominates inefficiency* in Retail (TCRR 65–89%) and Banking (58–72%), while Airline (new) is much lower (14–24%). (2) *Tool correctness is typically high* ($\geq 95\%$) across settings; on Airline-old for Gemini it is 98.58% with full parameter accuracy. (3) *Goal-shift recovery is strong* for GPT-4o and Sonnet on new sets (Airline 79–92%, Retail 88-90%), but substantially weaker for Gemini on new sets.

## E.2    PERSONA COVERAGE

Table 4: Persona coverage: number of tasks per persona.

| Persona | # Tasks |
|---------|---------|
| EASY_1 | 33 |
| EASY_2 | 34 |
| MEDIUM_1 | 69 |
| MEDIUM_2 | 34 |
| HARD_1 | 31 |

## E.3    FULL PER-MODEL METRICS (TOPLINE)

Table 5 reports per-domain results by model and set. We show overall success (TSR) alongside the three channels that compose TSR: communication (CI), actions, and NL assertions. Two patterns stand out: (i) *Airline (new)* keeps TSR respectable despite harder goal-shifted tasks because NL assertions stay high; (ii) *Retail (new)* for GPT-4o drops primarily via the communication channel (CI), even though the actions channel remains solid.

## E.4    EFFICIENCY, REDUNDANCY, AND RECOVERY

Table 6 decomposes efficiency (TUE with tool correctness and parameter accuracy), redundancy (overall TCRR with window & batch components), and adaptation (GSRT with counts of shifts, recovery rate, and transfers). Notably, Retail-new shows extreme redundancy across models (e.g., GPT-4o 89.14%), implying repeated lookups despite near-perfect parameter accuracy. Airline-new achieves low redundancy (13–24%) while maintaining high recovery for GPT-4o and Sonnet (79–92%). GSRT is not reported for Airline-new with Gemini-2.5-Flash due to insufficient credits to assess the recovery simulations.

## E.5    TUE ANALYSIS

With PA effectively at ceiling (mean $0.986$, $98.6\% \geq 0.95$), observed differences in TUE are driven by TC; the across-task TC box in Fig. **??** exposes long tails that a single averaged TUE score would otherwise hide.

Table 5: Topline metrics by domain/model/set. CI = Communicate Info channel.

| Domain | Set | Model | TSR (%) | CI (%) | Actions (%) | NL (%) |
|--------|-----|-------|---------|--------|-------------|--------|
| Airline | Old | GPT-4o | 64.84 | 27.78 | 65.18 | 60.79 |
| Airline | New | GPT-4o | 59.53 | 41.78 | 58.08 | 76.50 |
| Banking | — | GPT-4o | 51.25 | 28.34 | 51.17 | 70.31 |
| Retail | New | GPT-4o | 50.68 | 11.44 | 63.00 | 64.92 |
| Retail | Old | GPT-4o | 62.28 | 58.04 | 63.31 | 56.25 |
| Airline | New | Gemini-2.5-Flash | 40.74 | 29.33 | 44.09 | 45.22 |
| Airline | Old | Gemini-2.5-Flash | 53.98 | 22.22 | 49.91 | 62.58 |
| Banking | — | Gemini-2.5-Flash | 27.85 | 7.54 | 25.78 | 47.79 |
| Retail | New | Gemini-2.5-Flash | 51.26 | 14.38 | 63.80 | 63.18 |
| Retail | Old | Gemini-2.5-Flash | 64.80 | 66.06 | 64.77 | 72.92 |
| Airline | New | Claude-3.7-Sonnet | 69.90 | 61.56 | 66.92 | 81.33 |
| Airline | Old | Claude-3.7-Sonnet | 60.38 | 29.63 | 70.05 | 55.16 |
| Banking | — | Claude-3.7-Sonnet | 57.54 | 34.86 | 61.61 | 69.59 |
| Retail | New | Claude-3.7-Sonnet | 61.58 | 14.71 | 74.47 | 84.31 |
| Retail | Old | Claude-3.7-Sonnet | 79.57 | 79.12 | 79.66 | 81.25 |

Table 6: Efficiency and recovery. TUE reported as overall (ToolCorrectness / ParamAccuracy). TCRR as overall (Window / Batch). GSRT as (GoalShifts / Recovery% / Transfer%). "—" indicates not applicable.

| Domain | Set | Model | TUE (%)(TC/PA) | TCRR (%)(W/B) | Redun./Calls | GSRT (Shifts/Rec/Trans) |
|--------|-----|-------|----------------|---------------|--------------|-------------------------|
| Airline | Old | GPT-4o | 97.31 (95.52/100.00) | 36.57 (18.86/17.71) | 384/1050 | 179 / 91.6 / 8.4 |
| Airline | New | GPT-4o | 99.69 (99.48/100.00) | 13.54 (11.07/2.48) | 339/2503 | 90 / 92.2 / 7.8 |
| Banking | — | GPT-4o | 95.38 (92.31/100.00) | 61.54 (34.71/26.83) | 328/533 | 140 / 79.3 / 20.7 |
| Retail | New | GPT-4o | 98.82 (98.04/100.00) | 89.14 (57.77/31.37) | 591/663 | 50 / 88.0 / 12.0 |
| Retail | Old | GPT-4o | 97.29 (95.48/100.00) | 70.18 (44.42/25.76) | 1523/2170 | 258 / 91.9 / 4.3 |
| Airline | New | Gemini-2.5-Flash | 99.64 (99.40/100.00) | 17.63 (12.35/5.28) | 147/834 | — / — / — |
| Airline | Old | Gemini-2.5-Flash | 99.15 (98.58/100.00) | 14.46 (10.08/4.38) | 132/913 | 28 / 32.1 / 67.9 |
| Banking | — | Gemini-2.5-Flash | 98.93 (98.21/100.00) | 58.71 (27.90/30.80) | 263/448 | 123 / 57.7 / 41.5 |
| Retail | New | Gemini-2.5-Flash | 98.53 (97.55/100.00) | 66.45 (36.66/29.79) | 406/611 | 71 / 53.5 / 45.1 |
| Retail | Old | Gemini-2.5-Flash | 97.71 (96.18/100.00) | 68.70 (46.87/21.83) | 1545/2249 | 227 / 67.8 / 31.3 |
| Airline | New | Claude-3.7-Sonnet | 98.93 (98.21/100.00) | 24.11 (15.48/8.63) | 324/1344 | 101 / 79.2 / 19.8 |
| Airline | Old | Claude-3.7-Sonnet | 97.64 (96.07/100.00) | 36.73 (23.34/13.39) | 598/1628 | 227 / 90.7 / 8.4 |
| Banking | — | Claude-3.7-Sonnet | 95.34 (92.23/100.00) | 71.81 (42.59/29.22) | 693/965 | 142 / 58.5 / 40.1 |
| Retail | New | Claude-3.7-Sonnet | 97.74 (96.24/100.00) | 75.85 (44.08/31.77) | 807/1064 | 134 / 91.8 / 6.7 |
| Retail | Old | Claude-3.7-Sonnet | 98.18 (96.96/100.00) | 65.38 (40.38/25.00) | 1441/2204 | 324 / 89.5 / 9.0 |

