# OpenReview forum: "AgentChangeBench: A Multi-Dimensional Evaluation Framework for Goal-Shift Robustness in Conversational AI"
_ICLR.cc/2026/Conference — ICLR 2026 Conference Desk Rejected Submission_

### Official Review · Reviewer_kDAv · 2025-10-31

**Soundness:** 3
**Presentation:** 3
**Contribution:** 3
**Rating:** 2
**Confidence:** 5

**Summary:**

This paper presents a new agentic benchmark, focusing on goal shift.

**Strengths:**

It is a correct observation that other established benchmarks focus on static goals in a conversation, and they follow a golden path towards that goal. This is unrealistic for many real life applications. The paper proposes a much more realistic benchmark.

**Weaknesses:**

1. It is unusual for a benchmark paper not to show any results using the existing models. Why did you put all the results in Appendix? As is, it does not read like a paper. This is the biggest weakness. Conclusions mention some experiments but they do not exist in the main paper. I suspected whether I am reading a draft version of the paper.
2. On top of this, I'd have preferred at least a baseline approach for the authors to tackle the goal shift during the conversation.

**Questions:**

No questions

---

### Official Review · Reviewer_iMah · 2025-11-02

**Soundness:** 3
**Presentation:** 3
**Contribution:** 2
**Rating:** 2
**Confidence:** 3

**Summary:**

The paper addresses a critical gap in current benchmarks for conversational agents (LLM agents): most existing evaluations assume static user goals throughout the dialogue, overlooking the frequent goal changes that occur in real-world applications. The motivation is well-justified, as dynamic goal shifts are common in practical scenarios such as banking, retail, airline, and education. Agents must be able to detect, adapt to, and recover from these changes to be truly effective.

Main Contributions:

1. Novel Evaluation Dimension: This work is the first to systematically test LLM agents’ adaptability to mid-dialogue goal changes.
2 . Task Coverage: The authors construct 590 multi-turn tasks spanning four domains and five user personas, each with explicit goal shifts.
3. Methodological Framework: The paper introduces four complementary evaluation metrics: Task Success Rate (TSR), Tool Use Efficiency (TUE), Tool Call Redundancy Rate (TCRR), and Goal-Shift Recovery Time (GSRT).
4. Empirical Study: A cross-model evaluation reveals significant differences in adaptability and efficiency that are not captured by traditional pass@k metrics.

**Strengths:**

1. Multi-Domain Coverage: The benchmark covers four major real-world domains (banking, airline, retail, education), with diverse task types that align with enterprise needs.
2. Multi-Turn Dialogue & Goal Shift: By emphasizing multi-turn tasks and explicit goal shifts, the benchmark simulates realistic business workflows where user needs evolve during the conversation. This is more challenging and meaningful than traditional single-turn tasks.
3. Detailed Persona Design: Although the personas are relatively mild, the paper offers a nuanced segmentation of user behaviors (e.g., polite, curious, anxious, efficiency-oriented). This facilitates a more granular analysis of agent performance across different user preferences.

**Weaknesses:**

1. Incremental Advancement: The benchmark is primarily an extension of existing tool-use benchmarks, adding goal shift and persona dimensions. It lacks exploration of more advanced topics such as autonomous agent planning, multi-agent collaboration, and automatic goal recognition.
2. Lack of Extreme Scenarios: The benchmark does not systematically test for edge cases such as security boundaries, exception flows, or adversarial attacks.
3. Inconsistent Task Counts: The paper inconsistently reports the total number of tasks (590 vs. 565), and table statistics (e.g., Table 4) do not match the descriptions, which may confuse readers and cast doubt on the benchmark’s scale and coverage.
4. Statistical Ambiguity: Several tables lack clear definitions of their scope (e.g., domain, model, experimental set), making it difficult to interpret the data.

**Questions:**

NA

---

### Official Review · Reviewer_96FC · 2025-11-06

**Soundness:** 1
**Presentation:** 1
**Contribution:** 2
**Rating:** 2
**Confidence:** 4

**Summary:**

This work focuses on an important and practical problem: in real-world agent settings, users tend to shift their goals during interactions with agents. To evaluate how well agents can adapt to such goal shifts, this work introduces AgentChangeBench and proposes four complementary metrics to assess agents’ adaptability. Experimental results show that strong accuracy does not necessarily imply robustness under dynamic goals.

**Strengths:**

1. The motivation is both important and insightful, as in real human-agent interactions, users indeed tend to shift their goals.
2. The work is easy to follow, and the proposed benchmark provides a valuable foundation for future research in this area.

**Weaknesses:**

1. Motivation. The authors should elaborate further on the motivation for introducing persona. From my perspective, the main focus of this paper is on goal shifting, and the use of persona appears to serve the purpose of making the benchmark more realistic. A more detailed explanation of this design choice is necessary.
2. Experiments. The authors present one main experiment in the appendix; however, several questions remain.
(1) How do open-source models such as the Qwen3-series and GPT-OSS perform on AgentChangeBench?
(2) How do recently released large reasoning models (e.g., OpenAI o4-mini, DeepSeek-R1) perform?
(3) The authors mention pass^k; how does model performance vary across different values of k?
3. Writing. The main experiment should be placed in the main text for better readability. In addition, some important information, such as the model used to simulate the user and details like the temperature setting, is missing.

**Questions:**

1. In the abstract, the authors mention pass@k scores, but in the main text only pass^k appears. Further clarification or additional experiments are needed.
2. There are several relevant works on human–agent interaction, such as [1,2]; the authors are encouraged to cite and discuss them in the related work section.
3. In line 144, the authors state that *Transitions are triggered naturally.* How is naturally ensured? Is there any human judgment involved in verifying that the goal shifts occur naturally?
4. The authors should include an error analysis to better understand agents’ weaknesses in goal-shift scenarios, which could guide future improvements in this direction.

[1] LLM-Based Human-Agent Collaboration and Interaction Systems: A Survey
[2] CollabLLM: From Passive Responders to Active Collaborators

---

### Official Review · Reviewer_W6tz · 2025-11-07

**Soundness:** 2
**Presentation:** 3
**Contribution:** 2
**Rating:** 4
**Confidence:** 3

**Summary:**

This paper introduces AgentChangeBench, a novel benchmark designed to evaluate Large Language Model (LLM)-based conversational agents in scenarios where user goals shift dynamically during a multi-turn interaction. The authors correctly identify a significant gap in existing benchmarks (e.g., τ-bench, AgentBench), which typically assume static user objectives. The benchmark comprises 590 tasks across four domains (banking, retail, airline, education), incorporates five distinct user personas, and features explicit, annotated goal sequences. Beyond traditional success-rate metrics, the paper proposes a multi-dimensional evaluation framework including Task Success Rate (TSR), Tool Usage Efficiency (TUE), Tool Call Redundancy Rate (TCRR), and Goal Shift Recovery Time (GSRT). An empirical study across three major model families (GPT-4o, Gemini-2.5-Flash, Claude-3.7-Sonnet) demonstrates that this framework reveals performance trade-offs that a single metric would miss.

**Strengths:**

Novel and Relevant Contribution: The core idea—evaluating agents on their ability to handle mid-conversation goal shifts—is highly relevant and addresses a critical shortcoming in current evaluation paradigms. This focus directly improves the realism of agent assessment for enterprise deployment.
Comprehensive and Systematic Benchmark Design: The benchmark is well-constructed, with substantial scale (590 tasks), coverage across multiple realistic domains, and a clear, declarative task schema. The integration of five distinct personas adds a valuable layer of complexity and realism.
Multi-Dimensional Evaluation Framework: The proposed metrics (TSR, TUE, TCRR, GSRT) are a significant strength. They move beyond binary success/failure to provide insights into operational efficiency, cost (via redundancy), and adaptability, which are crucial for real-world application.
Rigorous Empirical Evaluation: The cross-model evaluation is thorough, testing three state-of-the-art models. The results convincingly demonstrate the benchmark's utility by surfacing clear differences in model performance across the various dimensions (e.g., Claude's fast recovery vs. Gemini's banking struggles vs. GPT-4o's balance).
Clarity and Reproducibility: The paper is generally well-written and structured. The inclusion of a detailed appendix with task examples, persona definitions, tool specifications, and full results supports reproducibility and transparency.

**Weaknesses:**

Statistical Reporting and Interpretation:
Lack of Statistical Significance: The results are presented as point estimates (e.g., TSR percentages) without any measures of variance or statistical significance testing (e.g., confidence intervals, p-values). Given that each task was run only 3 times, the stability of these metrics is unclear. Claims about model superiority (e.g., "Claude-3.7-Sonnet recovers fastest") would be significantly strengthened by statistical validation.
Inconsistent Precision: Some percentages are reported with two decimal places (e.g., 98.58%), while others are whole numbers or one decimal place. This can give a false impression of precision. A consistent reporting standard should be applied.
Interpretation of "Redundancy": The very high TCRR values (e.g., 89.14% for GPT-4o in Retail-new) are noted but not deeply interpreted. Is this a failure of the agent, a limitation of the tool design, or an inherent property of the retail domain? A brief discussion on the root cause of this observed redundancy would be valuable.
Writing and Presentation Issues:
Section Ordering: The paper currently jumps from the Introduction to Section 4 ("Empirical study") before covering Related Work (Section 2) and Methodology (Section 3). This is confusing and disrupts the narrative flow. The standard order (1. Intro, 2. Related Work, 3. Method, 4. Experiments) should be restored.
Clarity on "New" vs. "Old" Sets: The distinction between "new" and "old" task sets (e.g., Airline-new vs. Airline-old) is critical to understanding the results but is not explicitly defined in the main text. The reader must infer from Section 3.2 that "old" refers to adapted tasks from τ²-bench, while "new" are the contributions of this work. This should be clarified upfront in the methodology or experiment setup.
Conceptual Limitations (Acknowledged by Authors):
The authors rightly acknowledge limitations, such as the "benign" nature of the personas and the explicit (not implicit) nature of goal shifts. While acknowledged, these points remain weaknesses of the current benchmark version, as they limit its ability to test the full spectrum of adversarial or subtle real-world interactions.

**Questions:**

Statistical Significance: Given the relatively low number of runs per task (n=3), what is the estimated variance of your key metrics (TSR, GSRT)? Have you conducted any statistical tests to confirm that the performance differences between models are significant?
Metric Weighting: The weights for TSR (Communicate Info: 0.25, Action: 0.45, NL Assertion: 0.30) and TUE (Tool Correctness: 0.6, Param Accuracy: 0.4) are presented as determined by "empirical analysis" and "operational cost analysis." Could you provide more detail on this analysis? Was it domain-specific, and how sensitive are the overall results to these chosen weights?
Generalization and Scale: The benchmark focuses on customer service domains with structured APIs. How do you envision this framework generalizing to more open-ended domains (e.g., creative writing assistance, complex research synthesis) where goals are fuzzier and shifts are even more implicit?
Root Cause of Redundancy: The extremely high TCRR in some domains, particularly Retail, is a striking finding. What is your hypothesis for why this occurs? Is it an agent reasoning error, or could the benchmark's task design or toolset inadvertently encourage redundant calls?

---

### Note · Program_Chairs · 2026-01-17
**Submission Desk Rejected by Program Chairs**

The following references in this submission do not refer to real documents and/or have major errors in bibliographic information:

 Patil et al., “Gorilla: Large language model connected with massive APIs,” arXiv:2305.15334 (2023).
Zhou et al., “AppAgent: Multimodal agents for operating mobile apps,” arXiv:2308.00676 (2023).